# Non-Invasive Prediction of *IDH* Mutation in Patients with Glioma WHO II/III/IV Based on F-18-FET PET-Guided In Vivo ^1^H-Magnetic Resonance Spectroscopy and Machine Learning

**DOI:** 10.3390/cancers12113406

**Published:** 2020-11-17

**Authors:** Elisabeth Bumes, Fro-Philip Wirtz, Claudia Fellner, Jirka Grosse, Dirk Hellwig, Peter J. Oefner, Martina Häckl, Ralf Linker, Martin Proescholdt, Nils Ole Schmidt, Markus J. Riemenschneider, Claudia Samol, Katharina Rosengarth, Christina Wendl, Peter Hau, Wolfram Gronwald, Markus Hutterer

**Affiliations:** 1Department of Neurology and Wilhelm Sander-NeuroOncology Unit, Regensburg University Hospital, 93053 Regensburg, Germany; ralf.linker@ukr.de (R.L.); peter.hau@ukr.de (P.H.); markus.hutterer@ukr.de (M.H.); 2Institute of Functional Genomics, University of Regensburg, 93053 Regensburg, Germany; Fro-Philip.Wirtz@stud.uni-regensburg.de (F.-P.W.); peter.oefner@ukr.de (P.J.O.); Martina.Haeckl@stud.uni-regensburg.de (M.H.); claudia.samol@ukr.de (C.S.); wolfram.gronwald@ukr.de (W.G.); 3Department of Radiology and Division of Neuroradiology, Regensburg University Hospital, 93053 Regensburg, Germany; claudia.fellner@ukr.de (C.F.); christina.wendl@ukr.de (C.W.); 4Department of Nuclear Medicine, Regensburg University Hospital, 93053 Regensburg, Germany; jirka.grosse@ukr.de (J.G.); dirk.hellwig@ukr.de (D.H.); 5Department of Neurosurgery, Regensburg University Hospital, 93053 Regensburg, Germany; martin.proescholdt@ukr.de (M.P.); nils-ole.schmidt@ukr.de (N.O.S.); katharina.rosengarth@ukr.de (K.R.); 6Department of Neuropathology, Regensburg University Hospital, 93053 Regensburg, Germany; markus.riemenschneider@ukr.de; 7Department of Neurology, Saint John of God Hospital Linz, 4021 Linz, Austria

**Keywords:** glioma, *IDH* mutation, ^18^F-FET, ^1^H-MRS, D-2-hydroxyglutarate, linear support vector machine, glycine, myo-inositol

## Abstract

**Simple Summary:**

Approximately 75–80% of according to the classification of world health organization (WHO) grade II and III gliomas are characterized by a mutation of the *isocitrate dehydrogenase* (*IDH*) enzymes, which are very important in glioma cell metabolism. Patients with *IDH* mutated glioma have a significantly better prognosis than patients with *IDH* wildtype status, typically seen in glioblastoma WHO grade IV. Here we used a prospective O-(2-^18^F-fluoroethyl)-L-tyrosine (^18^F-FET) positron emission tomography guided single-voxel ^1^H-magnetic resonance spectroscopy approach to predict the *IDH* status before surgery. Finally, 34 patients were included in this neuroimaging study, of whom eight had additionally tissue analysis. Using a machine learning technique, we predicted *IDH* status with an accuracy of 88.2%, a sensitivity of 95.5% and a specificity of 75.0%. It was newly recognized, that two metabolites (myo-inositol and glycine) have a particularly important role in the determination of the *IDH* status.

**Abstract:**

*Isocitrate dehydrogenase* (*IDH)-1* mutation is an important prognostic factor and a potential therapeutic target in glioma. Immunohistological and molecular diagnosis of *IDH* mutation status is invasive. To avoid tumor biopsy, dedicated spectroscopic techniques have been proposed to detect D-2-hydroxyglutarate (2-HG), the main metabolite of *IDH*, directly in vivo. However, these methods are technically challenging and not broadly available. Therefore, we explored the use of machine learning for the non-invasive, inexpensive and fast diagnosis of *IDH* status in standard ^1^H-magnetic resonance spectroscopy (^1^H-MRS). To this end, 30 of 34 consecutive patients with known or suspected glioma WHO grade II-IV were subjected to metabolic positron emission tomography (PET) imaging with O-(2-^18^F-fluoroethyl)-L-tyrosine (^18^F-FET) for optimized voxel placement in ^1^H-MRS. Routine ^1^H-magnetic resonance (^1^H-MR) spectra of tumor and contralateral healthy brain regions were acquired on a 3 Tesla magnetic resonance (3T-MR) scanner, prior to surgical tumor resection and molecular analysis of *IDH* status. Since 2-HG spectral signals were too overlapped for reliable discrimination of *IDH* mutated (*IDHmut*) and *IDH* wild-type (*IDHwt*) glioma, we used a nested cross-validation approach, whereby we trained a linear support vector machine (SVM) on the complete spectral information of the ^1^H-MRS data to predict *IDH* status. Using this approach, we predicted *IDH* status with an accuracy of 88.2%, a sensitivity of 95.5% (95% CI, 77.2–99.9%) and a specificity of 75.0% (95% CI, 42.9–94.5%), respectively. The area under the curve (AUC) amounted to 0.83. Subsequent ex vivo ^1^H-nuclear magnetic resonance (^1^H-NMR) measurements performed on metabolite extracts of resected tumor material (eight specimens) revealed myo-inositol (M-ins) and glycine (Gly) to be the major discriminators of *IDH* status. We conclude that our approach allows a reliable, non-invasive, fast and cost-effective prediction of *IDH* status in a standard clinical setting.

## 1. Introduction

Gliomas are extremely aggressive brain tumors associated with a poor median overall survival (between 15 and 26 months in glioblastoma (GBM) patients) [1,2].

These tumors are classified by the use of a multi-layer classification including histopathological and molecular features andgrading according to the world health organization (WHO). The isocitrate dehydrogenase (*IDH*) status has a special meaning in this classification system [3]. Patients with WHO grade II and III glioma typically harbor *IDH* mutations in about 75% to 80% of cases [4,5], whereas patients with primary GBM in general do not [3]. The *IDH* status improves the discrimination of prognosis in comparison to the WHO grade alone [4,6,7]. As a result, tumor metabolism in *IDH* mutated glioma cells has come to the fore as an important diagnostic and therapeutic target [6]. Following this concept, the whole metabolism of a respective brain tumor and not only specific substrates, adapts as a consequence of *IDH* mutation [8,9,10,11,12,13]. Consecutively, *IDH* wildtype astrocytoma of WHO grade II and III is a provisional entity that is strongly discouraged. Diffuse astrocytic glioma *IDH* wild-type *(IDHwt*) presenting with disadvantageous genetic characteristics (e.g., epidermal growth factor receptor gene (EGFR) amplification or combined chromosome 7 gain/chromosome 10 loss or telomerase reverse transcriptase (TERT) promoter mutation) was proposed to be a “diffuse astrocytic glioma, *IDHwt*, with molecular features of glioblastoma” [14] and will be termed glioblastoma in the upcoming revised 2021 classification. 

Three *IDH* isoforms have been described so far. *IDH-1* is localized in peroxisomes and the cytoplasm, whereas *IDH-2* and *IDH-3* are part of the tricarboxylic acid cycle (TCA) in mitochondria. All three isoforms convert isocitrate to α-ketoglutarate (α-KG) by oxidative decarboxylation [15]. To date, in human gliomas, only heterozygotic mutations in *IDH-1* and *IDH-2* have been reported, with a significant preponderance of *IDH-1* [6,16]. Around 90% of *IDH-1* mutated (*IDHmut*) gliomas involve a substitution in the catalytic site of arginine-132, which is replaced by histidine (*IDH1 R132H* mutation) [6]. As only one allele is mutated, the wild-type allele will continue to produce α-KG, while the neomorphic enzyme activity conferred upon *IDHmut* will catalyze the conversion of α-KG to the oncometabolite D-2-hydroxyglutarate (2-HG) [17], which is measurable in vitro and in vivo. 

To detect *IDH* mutations, neuropathological and imaging techniques have been developed. Immunohistochemistry (IHC) for *IDH1-R132H* [18] and, if negative, panel sequencing (to detect the rarer mutational variants, at least in patients below 55 years and cases with negative IHC) are regarded as gold standard [3,19]. However, both methods depend on the availability of tumor tissue. Non-invasive proton magnetic resonance spectroscopy (MRS) holds great potential in the routine analysis of 2-HG [20], especially in eloquent areas of the brain with increased surgical risk as well as in the monitoring of *IDH*-associated metabolites during potential *IDH*-specific therapies [21,22,23]. So far, different MRS techniques have been used to predict the *IDH* mutation status in brain tumors but the best results were reported for direct determination of 2-HG by MRS [24]. Other studies have attempted to infer the *IDH* mutational status indirectly by applying machine learning methods to magnetic resonance imaging (MRI) patterns [25]. Alternatively, O-(2-^18^F-fluoroethyl)-L-tyrosine (^18^F-FET) uptake parameters and textural features in positron emission tomography (PET) have been exploited for the determination of *IDH* mutation status [26,27,28]. So far, ^18^F-FET uptake has not been used in combination with ^1^H-MRS to optimize voxel placement in diffuse glioma. 

In view of the technical challenges of earlier studies and with the aim to develop a reliable, prospective, technically feasible, fast and affordable approach to discriminate *IDH* wild type and *IDH* mutated gliomas in vivo, we aimed here to process ^18^F-FET PET-guided routine ^1^H-MRS data from a standard 3T-MR scanner in a fully automatic manner to reveal *IDH* mutations with high accuracy. 

## 2. Results

### 2.1. Patient Characteristics

Thirty-four patients with glioma WHO grade II/III/IV and known *IDH* mutation status were prospectively included in this investigator-initiated, cross-sectional, monocentric study. Detailed patient characteristics are given in Table 1. Twenty-two patients harbored an *R132H IDH-1* mutation verified by IHC (*IDHmut* group). No *IDH-2* mutation was detected. Twelve patients suffered from *IDHwt* glioma (*IDHwt* group) corroborated by genomic sequence analysis in tumor tissue of all *IDHwt* patients. Tumor tissue of 7 *IDHmut* patients harbored a combined loss of heterozygosity (LOH) 1p/19q and was, therefore, classified as an oligodendroglial tumor (32% of *IDHmut* patients). In glioma with *IDH* mutation, a higher rate of O6-Methylguanine-DNA-Methyltransferase (MGMT) promotor methylation was detected than in *IDHwt* glioma.

Age at first diagnosis and at study inclusion was considerably lower in the *IDHmut* in comparison to the *IDHwt* group. The proportion of patients included in first line setting versus relapse was comparable in both groups (first-line setting *IDHmut* 64% versus *IDHwt* 50%).

### 2.2. Standard ^1^H-MRS at 3T Is Not Sufficient to Reliably Detect 2-HG

^1^H-MRS in a voxel of 14 × 14 × 14 mm was acquired as described below. Typical ^1^H-MR spectra of a patient with anaplastic astrocytoma, *IDHmut*, WHO grade III are shown in Figure 1 (1a: healthy brain region, 1b: tumor region). Typical tumor metabolites, for example, increased lactic acid (Lac) and choline (Cho) and decreased N-acetylaspartate (NAA), were detected as expected for glioma. An unambiguous detection of a 2-HG peak employing standard ^1^H-MRS was not feasible, as different components of the 2-HG signal (two Hβs at 1.91 ppm and two Hγs at 2.24 ppm) overlapped with signals of glutamate (Glu), glutamine (Gln) and γ-aminobutyric acid (GABA), while the signal of the Hα proton overlapped with numerous compounds such as myo-inositol (M-ins). Figure 1b demonstrates that the 2-HG peak at 2.24 ppm is hardly distinguishable from the Glu/Gln peaks. Consequently, predicting *IDH* mutation in glioma solely based on the 2-HG signal without the use of sophisticated pulse sequences specific for 2-HG detection might be unreliable.

### 2.3. IDH Mutation Induces a Specific Pattern of Alterations in ^1^H-MR Spectra

Based on the published data that *IDH* mutations induce a whole range of metabolic changes beyond the increase of 2-HG, we next investigated whether additional spectral information could be used to predict the *IDH* mutation status. For this purpose, we generated averaged spectra of *IDHmut* and *IDHwt* tumors to detect relevant spectral differences between the two groups (Figure 2). 

As can be seen the signal at 2.24 ppm corresponding to 2-HG was also difficult to detect in the averaged spectra of patients bearing an *IDH* mutation. In contrast, distinct spectral changes were visible between *IDHwt* and *IDHmut* tumors in the region between 4.1 ppm and 3.5 ppm, indicating that *IDHmut* may cause further metabolic changes in addition to the production of 2-HG in ^1^H-MR spectra. Therefore, we decided to analyze the complete spectral information for predicting the *IDH* mutation status.

### 2.4. A Linear Support Vector Machine Provides High Sensitivity and Specificity in Detecting IDHmut

Next, a linear support vector machine (SVM) was used for the prediction of *IDH* mutations based on ^1^H-MR spectra. To this end, a nested leave-one-out cross-validation method was applied. In this approach the set of all patient spectra was iteratively split into a training set and a test set. In each iteration, the classification algorithm was trained on the training set. Next, the trained algorithm was employed to predict the mutation status of the test set. By this, it was ensured that the training step was not influenced by the test set, thus obtaining a practically unbiased assessment of the true classification error [29]. For a maximal prediction performance, the choice of predictive features was optimized in an inner cross-validation. This resulted on average in the selection of two predictive features, which allowed the prediction of an *IDH* mutation with an average accuracy of 88.2%, a specificity of 75.0% (95% CI, 42.9–94.5%) and a sensitivity of 95.5% (95% CI, 77.2–99.9%). The area under the ROC curve (AUC) amounted to 0.83 (Appendix A). Prediction results of all samples are provided in Appendix A. 

### 2.5. In Vivo and Ex Vivo Analysis of Metabolites

Lastly, the classification algorithm enabled us to focus on the discriminating features of high importance (green box in Figure 3, top). To get a more detailed view of the underlying discriminatory metabolites, hydrophilic extracts of excised tumor material (eight specimens) were analyzed ex vivo by high-resolution ^1^H nuclear magnetic resonance (NMR) spectroscopy. For the assignment of NMR signals to specific compounds, spectra were superimposed with reference spectra obtained from pure compounds. As visible from Figure 3, the features selected by the classification algorithm mostly corresponded to M-ins and glycine (Gly). 

Furthermore, a comparison of two inlays displayed in the lower part of Figure 3 clearly shows the 2-HG signal in the ex vivo high-resolution spectrum of an exemplary patient bearing an *IDH* mutation, while—as expected—this signal is absent in a patient without mutation. These details can only be seen in the high-resolution ex vivo spectra, while these features are invisible due to a substantially increased linewidth in the in vivo spectra. The complete data acquisition and analysis pipeline is summarized in Figure 4.

## 3. Discussion

Here, we performed a prospective, investigator-initiated, cross-sectional, monocentric study to evaluate the non-invasive prediction of *IDH* mutation status in glioma patients using standard single voxel ^1^H-MRS. Our results demonstrate that reliable non-invasive prediction of the *IDH* mutation status is feasible employing a combination of routine ^1^H-MRS at 3T together with data analysis using a machine learning approach. In addition, we show that in vivo ^1^H-MRS corresponds very well with ex vivo spectra of fresh frozen tumor specimens. Importantly, the signals of Gly and M-ins but as explained below not the signal of 2-HG, could be identified as discriminating features in in vivo ^1^H-MRS analyses in tumor tissues associated with *IDH* mutation. 

Numerous clinical trials predicting *IDH* mutation status using MRI/MRS/radiomics have been conducted with a summary sensitivity across all techniques of 86% (95% CI, 79–91%) and a summary specificity of 87% (95% CI, 91–100%), while pooled sensitivity and specificity for 2-HG MRS were 95% (95% CI, 85–98%) and 91% (95% CI, 83–96%), respectively [24]. Longer echo times (significant increased sensitivity and specificity in TE 97 ms compared to TE 30–35 ms) [30], ultra-high-field MRI (7T) [31,32], 2D correlation spectroscopy (COSY) at 3T [33] and 2D L-COSY at 7T [34] were used to improve detection of the 2-HG signal. Different post-processing techniques have also been applied in various settings, mostly in a limited number of patients (e.g., 38 patients in Reference [35]). Interestingly, independent of 2-HG, peak alterations of other metabolites were frequently seen (e.g., Cho, Glu, Gly, glutathione, cystathionine [36,37,38]) and sometimes used to improve detection of *IDH* mutation status, for example, combination of 2-HG and Glu levels [39]

The described techniques are frequently not suitable for clinical routine as they need extensive technical and personal capacities [24]. For example, ultra-high-field MRI (7T) [31,32] is only available at a few quaternary university hospitals but not in daily clinical practice. 

In comparison to the cited data, our approach presents a comparable prediction accuracy of 88.2%, sensitivity of 95.5% (95% CI, 77.2–99.9%) and specificity of 75.0% (95% CI, 42.9–94.5%). Furthermore, the acquisition of ^1^H-MR spectra and all subsequent steps are performed in a fully automated fashion. Therefore, time-consuming complex technologies were not necessary and ^1^H-MRS caused on average twenty minutes of additional examination period following standard MRI and was well accepted by patients. Moreover, ^18^F-FET PET imaging, which could be performed in 30/34 patients, facilitated optimal voxel placement in ^1^H-MRS and, thus, contributed to diagnostic accuracy. 

*IDHmut* glioma cells show alterations in several enzyme activities, resulting in various metabolic changes, including a distinct intra- and extracellular increase of 2-HG [6] and a decrease in Glu concentrations [11,40]. Therefore, our first study hypothesis was that the 2-HG peak at 2.24 ppm of the ^1^H-MR spectrum in vivo is associated with an *IDH* mutation. Unfortunately, peaks of 2-HG and Glu overlapped heavily in ^1^H-MR spectra at the position 2.24 ppm as seen in Figure 1 and Figure 2. Furthermore, only minimal changes in global intensity were seen in the presence of mutated *IDH*. Consequently, this feature was not applicable to prediction of *IDH* status in vivo. However, the region between 4.1 ppm and 3.5 ppm demonstrated distinct differences between *IDHwt* and *IDHmut* glioma ^1^H-MR spectra. For this reason, our next study objective aimed at the unbiased selection of features in whole single voxel ^1^H-MR spectra that were capable of discriminating *IDHwt* from *IDHmut* gliomas using a machine learning approach. Starting with 250 features per spectrum, we finally identified two discriminating features of high importance, corresponding to Gly and M-ins.

In Figure 2, the region of the two predictive features is indicated by a green box. Of note, the peak on the left, corresponding to Gly, is clearly decreased in *IDHmut* gliomas, while the opposite is true for the peak on the right, which corresponds to M-ins (Figure 3, bottom). In line with our results, Miyata et al. had also shown previously a clear reduction of Gly in *IDH*-mutated glioma [40]. On the other hand, increased Gly levels and Gly/2-HG ratios have been associated with a more aggressive type of glioma with rapid progression and shorter patient survival [37]. Non-essential amino acids such as Gly are generated from or degraded to intermediates of glycolysis and the TCA cycle. As *IDH* mutations affect the production and availability of α-KG and further downstream intermediates of the TCA cycle, the observed decrease in Gly level is not unexpected. 

A high M-ins level in ^1^H-MRS is relevant to distinguish glioma from other brain tumors, for example, primary central nervous system lymphomas [41]. Furthermore, *IDH* mutation seems to be a relevant factor for M-ins concentration. A positive correlation between 2-HG and M-ins has been reported recently [32] but the connecting link is missing so far. M-ins is an organic osmolyte in the brain and its intracellular concentration depends on the osmolality dominant in the surroundings. Therefore, increased M-ins concentration in *IDH* mutated gliomas may cause a change in metabolism of glioma cells following the osmotic alterations in the tumor microenvironment [42]. Moreover, in brain tumors a link between M-ins levels and the expression of the enzyme inositol 3-phosphate synthase (ISYNA1) has been reported [41]. ISYNA1 is responsible for biosynthesis of M-ins by metabolizing glucose-6-phosphate to inositol-1-phosphate. In summary, recapitulating the role of M-ins in *IDHmut* glioma is exciting and deserves further investigations. 

Despite the small number of patients and a missing coregistration of ^18^F-FET PET and anatomical MRI data, a relevant strength of the current study is the high accuracy in detecting *IDH* mutation status independent of the heterogenous population showing that our technique works in distinct settings, including patients with various neuropathological diagnoses and first-line or relapse setting with different treatment modalities in the past. Furthermore, our method is based on routine ^1^H-MRS measurements and a fully automated analysis pipeline making it amenable to routine clinical practice.

## 4. Materials and Methods

### 4.1. Patients Characteristics

Forty-three patients with known or suspected glioma WHO II to IV in a first diagnosis or relapse situation and planned tumor biopsy or resection were screened for the study in a single neuro-oncological center (University Hospital Regensburg, Regensburg, Germany) between December 2015 and September 2019. Nine patients (21%) were excluded from analysis, including a histology other than glioma (two patients, 5%), no surgery (three patients, 7%), not conducting ^1^H-MRS (two patients, 5%) and due to technical difficulties in ^1^H-MRS evaluation in two patients (5%). Consequently, thirty-four patients (80%) were enrolled (Appendix A). Clinical parameters, such as age, gender and Karnofsky Performance Score, were evaluated for all patients.

This prospective clinical trial was approved by the local ethics committee (Ethikkommission bei der Universität Regensburg, reference number 15-101-0258, approval date: 21 October 2015) and all patients gave written informed consent for the use of their clinical, molecular and imaging data.

### 4.2. Glioma Diagnosis and Molecular Parameters

The patients’ brain tumor diagnosis and WHO grading were determined at the local neuropathology department. Routine histopathology was accompanied by testing for *IDH* mutation (standard immunohistochemistry and Sanger sequencing) and *MGMT* promoter methylation status (methyQESD) [43,44]. Promoter methylation was considered as being present starting from a minimum of 3% methylation. Examination of LOH 1p/19q status was performed in all *IDHmut* gliomas identified by microsatellite analysis [45].

### 4.3. Standard MRI and ^18^F-FET PET Imaging

Standard MRI was routinely performed at the local department of radiology (University Hospital Regensburg) using a clinical 3T-MR scanner (Magnetom Skyra, Siemens Healthcare, Erlangen, Germany) and a 20-channel head-neck array coil. The routine MRI protocol included T1-weighted sequences with and without contrast agent (T1, T1wCE), T2 and FLAIR sequences and diffusion-weighted imaging (DWI) including the apparent diffusion coefficient (ADC).

^18^F-FET PET was carried out in a routine manner at the local department of nuclear medicine corresponding to the German and Austrian guidelines for brain tumor imaging with the use of labeled amino acid analogues [46]. In twenty-one patients PET scans were performed on a Biograph 16 PET/CT scanner, in nine patients a Biograph mCT40 scanner was used (both CTI-Siemens, Erlangen, Germany). In four patients no PET scans were available. Using ^18^Fluor from our on-site cyclotron ^18^F-FET was produced in-house as described previously [47] and administered according to the German Medicinal Products Act, AMG §13 2b and in accordance with the local regulatory authority (Regierung von Unterfranken). The need of a permission according to the radiation protection law was negated in response to an inquiry at the German Federal Office for Radiation Protection. All patients fasted for at least 6 h before PET scanning. Prior to the PET scan, a low-dose CT scan was carried out for attenuation correction. Three-dimensional PET was performed 5–15 min and 20–30 min after intravenous injection of 229 ± 44 MBq ^18^F-FET. The emission data set was attenuation-corrected using low-dose CT and reconstructed applying the OSEM (Ordered Subset Expectation Maximization) algorithm with parameters recommended by the manufacturer.

### 4.4. Data Acquisition of ^1^H-MRS 

In vivo single voxel ^1^H-MRS was performed on the same clinical 3T-MR scanner as used for the standard MRI scan in the same session. ^1^H-MRS voxels for acquisition of the tumor metabolism were placed into the most representative tumor areas identified by standard MRI (e.g., contrast-enhancement on T1-sequences, glioma-typical T2/FLAIR signal alterations) and ^18^F-FET PET (identification of the metabolic tumor volume and the area of maximal ^18^F-FET uptake) (Figure 4). In a visual procedure using anatomical landmarks and applying comparable angulation in ^18^F-FET PET and MRI/MRS the area of maximal ^18^F-FET uptake was integrated in voxel positioning. In 30 of 34 patients, ^18^F-FET PET was available for ^1^H-MRS voxel planning. Furthermore, tumor calcification (MRI and CT scan for attenuation correction during ^18^F-FET PET) as well as a close relationship of the tumor to cranial bones and to the ventricle with cerebrospinal fluid – known interference factors for ^1^H-MRS were considered.

All spectra were acquired using a PRESS (Point RESolved Spectroscopy) sequence with a voxel size of 14 × 14 × 14 mm, a TR of 2000 ms and a TE of 30 ms; 100 acquisitions were averaged for each spectrum. Acquisition time (excluding adjustment procedures) was 3 min 30 s. A spectral width of 1200 Hz centered at the residual water signal was used. Data were Fourier transformed to a final size of 1024 real data points.

### 4.5. ^1^H-MRS Data Pre-Processing

Data files from the 3T-MR scanner in Siemens “.rda” format were transferred to the clinical and biomedical MRS software package jMRUI [48]. Within jMRUI spectra were phase corrected and referenced with respect to the choline signal at 3.20 ppm. Next, all data points of each spectrum were exported for further analysis in “.txt” format. All further analyses were performed employing the statistical computing environment R v. 3.6.1 (Development Core Team 2009). To achieve comparability across data sets, all spectra were first automatically aligned with respect to the clearly visible choline signal. To account for remaining small variations in signal positions an equidistant binning was performed by combining four consecutive data points into one bin, which resulted in a total of 250 bins or features. Next, the remaining water signal and regions containing no signals were discarded so that only the features in the region between 4.5–0.7 ppm were kept for subsequent analyzes. Next for the reduction of unwanted sample-to-sample variations, all samples were normalized to a total signal intensity of 1, followed by a log_2_ transformation for the reduction of heteroscedasticity.

### 4.6. ^1^H-MRS Classification

^1^H-MRS classifications were obtained employing a linear support vector machine (SVM). To this end, the R-library e1071 (http://cran.r-project.org/web/packages/e1071/) was used. To guarantee an almost unbiased estimate of the true prognostication error [29], a nested leave-one-out cross-validation approach was used. In this procedure, iteratively one spectrum was selected for testing while the remaining spectra were used for training of the classification algorithm. In all cases the samples used for training and testing were strictly separated from each other, which is of prime importance for a realistic estimation of the true classification error. The inner loop of the nested cross-validation was employed to estimate the best number of features using a t-test based ranking of all features in the region between 4.5–0.7 ppm. The number of selected features was increased in steps of one from a starting value of one until optimal classification performance was obtained. Note that this feature selection step was performed on the training data only. The cost parameter was left on its default value of one. The complete R-script for preprocessing and sample classification is provided in the Appendix A. 

### 4.7. Ex Vivo Evaluation of Tumor Material

In total 8 samples of excised tumor tissue (5 *IDHmut* and 3 *IDHwt*) were analyzed. Neurosurgical biopsy or resection was done immediately after ^1^H-MRS (range 1–16 days).

Excised tumor material was snap frozen in liquid nitrogen and kept at −80 °C prior to measurement. To this end, tissue specimens of 40 to 170 mg, which were cooled on ice, were put in Precellys lysing kit tubes (Bertin Technologies, Montigny-le-Bretonneux, France), followed by the addition of 1 mL of 80% (*v*/*v*) aqueous MeOH and homogenization for 2 × 20 s at 6500 rpm. Next, as an extraction standard, 10 µL of 20 mM nicotinic acid were added. Resulting mixtures were centrifuged at 8960 *g* for 5 min at a temperature of 4 °C. Supernatants were collected and the remaining pellets were washed two times with 400 µL 80% (*v*/*v*) methanol followed by centrifugation of the first and second wash with 8960 *g* and 12,902 *g*, respectively. Next, the supernatants of each sample were combined and evaporated. For NMR measurements, dried remains were dissolved in 400 μL pure water, mixed with 200 μL of 0.1 mol/L phosphate buffer, pH 7.4 and 50 μL deuterium oxide containing 0.75% (*w*/*v*) 3-trimethylsilyl-2,2,3,3-tetradeuteropropionate (TSP)(Sigma-Aldrich, Taufkirchen, Germany). In this context TSP is serving as internal standard for referencing and quantification.

NMR measurements were conducted on a 14.1 T (600 MHz) Bruker Avance III NMR spectrometer (Bruker BioSpin GmbH, Rheinstetten, Germany). It was equipped with a triple-resonance (^1^H, ^13^C ^15^N, ^2^H lock) cryogenic probe with z-gradients. Furthermore, an automatic cooled sample changer was used. For each sample, a 1D ^1^H-NMR spectrum was carried out following established protocols [49]. As described previously, individual NMR signals were assigned to their respective metabolites by superposition with reference spectra of pure compounds [49].

## 5. Conclusions

We established a non-invasive, reliable and easy to practice method for prediction of *IDH* mutation status in glioma based on ^18^F-FET PET-guided standard ^1^H-MRS and machine learning techniques. Analysis was based on two discriminating features corresponding to Gly and M-ins but not on 2-HG, as described in the literature. We will initiate a larger in-house validation cohort and propose to perform an independent study to confirm our findings, using the same technical approach. When validated, application in clinical routine will be easy to perform. 

The R-code for all predictions performed in this study including an easy to use graphical interface is available upon request from the authors.

## Figures and Tables

**Figure 1 cancers-12-03406-f001:**
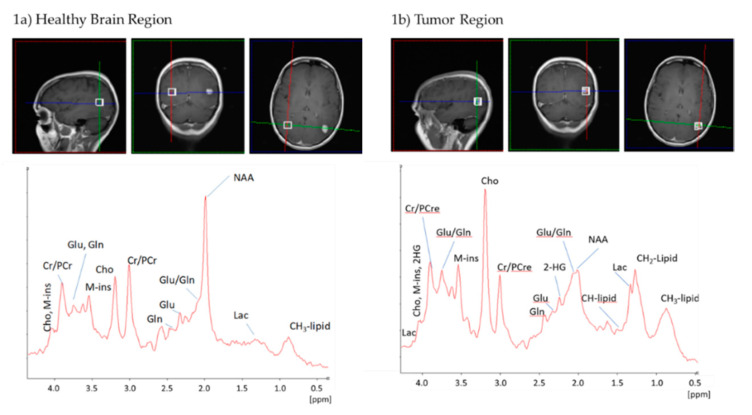
Exemplary ^1^H-magnetic resonance spectra (^1^H-MRS) of a glioma patient bearing an *IDH* mutation. Both spectra were acquired by a 3T MR scanner in a voxel of 14 × 14 × 14 mm. (**a**) healthy brain region (**b**) tumor region.Abbreviations: 2-HG, D-2-hydroxyglutatrate; CH2-Lipid, CH2 groups of lipids; CH3-Lipid, CH_3_ groups of lipids; Cho, choline; Cr, creatine; Gln, glutamine; Glu, glutamate; Lac, lactate; M-ins, myo-inositol; NAA, N-acetylaspartate; PCr, phosphocreatine.

**Figure 2 cancers-12-03406-f002:**
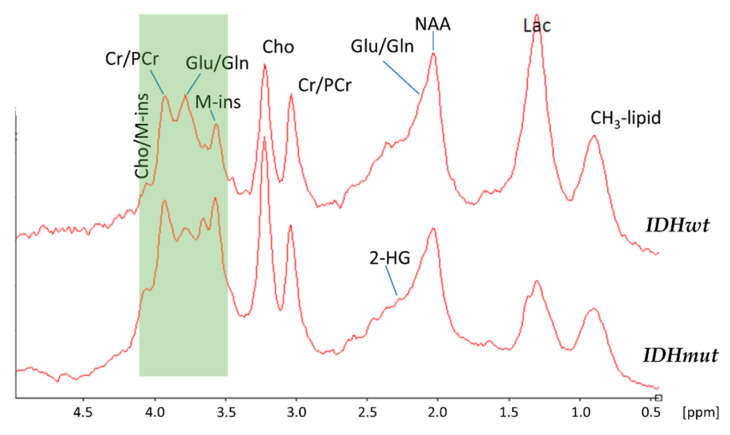
Averaged ^1^H-MR spectra of *IDHwt* (n = 9, top) and *IDHmut* tumors (n = 17, bottom). The region between 4.1 ppm and 3.5 ppm (colored green) demonstrated distinct spectral differences between *IDHwt* and *IDHmut*
^1^H-MR spectra.

**Figure 3 cancers-12-03406-f003:**
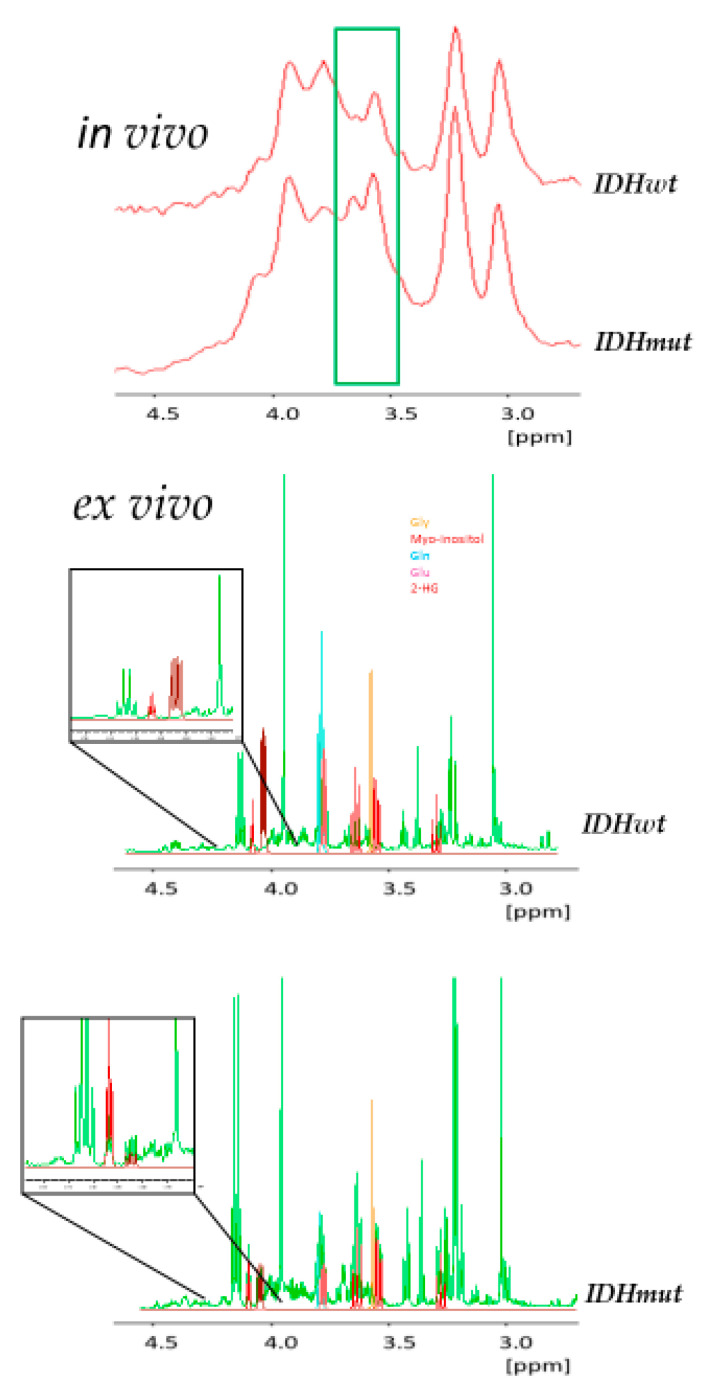
Predictive features. (top) Exemplary in vivo ^1^H-MR spectra of *IDHwt* and *IDHmut* glioma acquired at 3T. The region of the two features used for predicting *IDH* mutation status is marked in green. (bottom) Exemplary ex vivo high-resolution 1D ^1^H-nuclear magnetic resonance (NMR) spectra acquired at 14.1 T of methanol extractions of excised *IDHwt* and *IDHmut* glioma (green). For identification of metabolites pure compound spectra of Gly, M-ins, Gln, Glu and 2-HG are overlaid with these spectra. The two inlays show the region of the expected signal of 2-HG (dark red).

**Figure 4 cancers-12-03406-f004:**
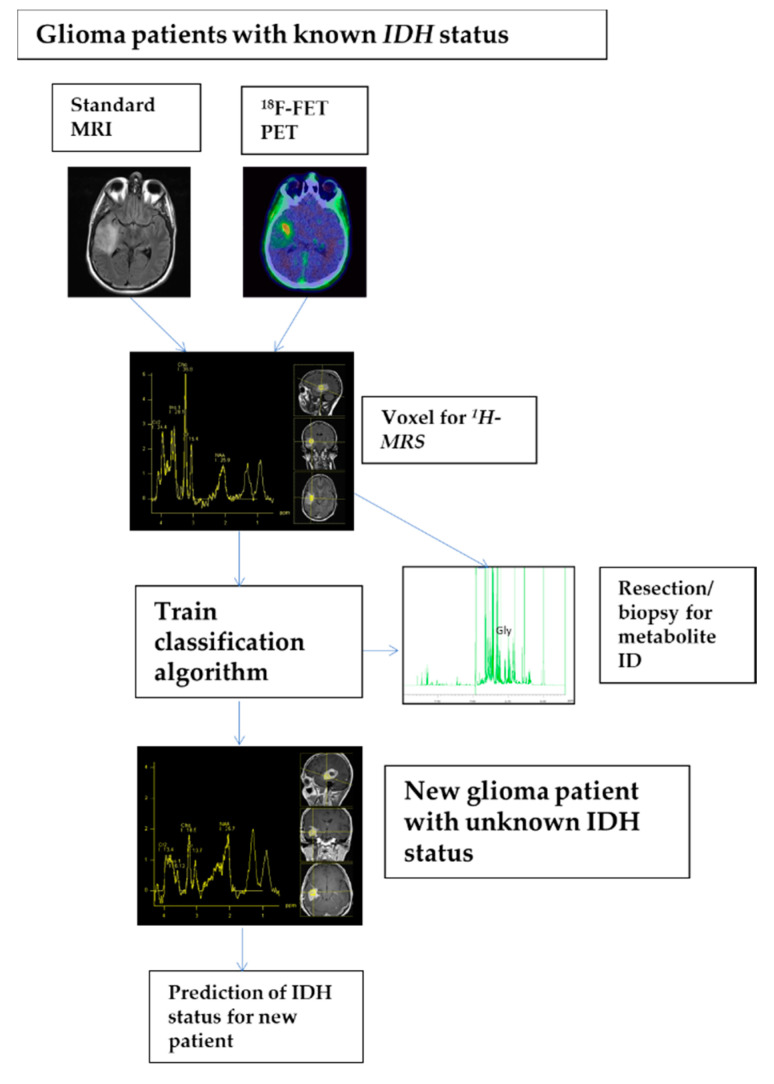
Detailed process description of analysis pipeline as it could be applied in clinical praxis. First, standard MRI and ^18^F-FET PET were applied for voxel planning. Second, in selected voxels 1D ^1^H MRS spectra of patients with known *IDH* status were acquired. Next, a classification algorithm, namely a linear support vector machine, was trained on all spectra of the training set implementing a nested leave-one-out cross-validation approach. Finally, once training is completed new spectra of patients with unknown *IDH* status may be presented to the algorithm for prediction of *IDH* status.

**Table 1 cancers-12-03406-t001:** Study patient characteristics dependent on *IDH* mutation status.

		*IDH1*-Mutation
		*IDHwt*	*IDHmut*
		Mean	Median	Number	in %	Mean	Median	Number	in %
Age at first diagnosis		55	54			39	37		
Age at begin of trial *		55	55			43	43		
Gender	male			6	50%			12	55%
female			6	50%			10	45%
Diagnosis	diffuse astrocytoma			0	0%			3	14%
oligodendroglioma			0	0%			4	18%
anaplastic astrocytoma			2	17%			9	41%
anaplastic oligodendroglioma			0	0%			4	18%
glioblastoma			10	83%			2	9%
WHO	II			0	0%			7	32%
III			2	17%			13	59%
IV			10	83%			2	9%
MGMT-Status	non-methylated			8	67%			6	27%
methylated			4	33%			15	68%
unknown			0	0%			1	5%
LOH1p19q	no			6	50%			14	64%
yes			0	0%			7	32%
unknown			6	50%			1	5%
Karnofsky	80			5	42%			0	0%
90			3	25%			5	23%
100			4	33%			17	77%
Timepoint of Inclusion	first-line treatment			6	50%			14	64%
first relapse			5	42%			4	18%
>first relapse			1	8%			4	18%
		* range 25–73 years		* range 21–66 years

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
