# Peer review of "Non-Invasive Prediction of IDH Mutation in Patients with Glioma WHO II/III/IV Based on F-18-FET PET-Guided In Vivo 1H-Magnetic Resonance Spectroscopy and Machine Learning"

_cancers, 2020, doi:10.3390/cancers12113406_

Round 1

Reviewer 1 Report

Review

In this study, the authors explored the use of single voxel 1H-MR spectroscopy in combination with machine learning for the non-invasive and fast diagnosis of IDH gene mutation status in 34 consecutive patients suffering from glioma.  Additionally, 18F-FET was used for optimized 1H-MRS voxel placement in 30 of the 34 patients. Their approach predicted IDH gene mutation status with similar accuracy compared to techniques that are not suitable for clinical routine. They found that myo-inositol and glycine were the major discriminators of IDH gene mutation status.

Non-invasive prediction of IDH gene mutation status of glioma is of high clinical relevance.

The manuscript is very well written and the results are presented in an understandable way. However, a few points need revision.

Abstract, line 40: Please mention that NMR was performed in eight patients only.

Introduction, line 54: The authors should include data for percentage of IDH mutation in glioma WHO° II and III, with references.

Introduction, line 54: The authors claim “… will be termed glioblastoma in the upcoming revised 2021 classification”. Is there a reference available for this statement?

Methods, line 281: FET PET was used to guide MRS voxel positioning. Was coregistration of FET PET data and anatomical MRI data performed prior to planning of the MRS experiment? The authors should provide more details about the workflow of FET PET guided MRS voxel positioning.

Methods, line 285: A fixed voxel size of 14 x 14 x 14 mm3 was used. Why the authors did not use individual voxel sizes adapted to the tumor size? This would help to reduce partial volume effects to peritumoral areas. Were all tumors large enough for this fixed voxel size? Please comment. A Table with the diameters of the tumors in all three directions of space might be informative.

Methods, line 286: Which sequence scheme was used for the MRS experiment? A PRESS sequence?

Methods, line 297: The preprocessing of the spectra is not sufficiently described. Did the authors perform phase or frequency shift correction? Did they perform spectral fitting (time or frequency domain)? Did they use the fit for the further data analysis or the original spectra?

Methods, line 304: Which ppm region did “contain no signal”?

Methods, line 312: The authors used a nested leave-one-out cross-validation approach. I am not sure and I am not an expert, but I wonder if there is a correction for multiple comparisons necessary? Furthermore, each data set was used for training and for testing, is this correct? In my opinion, there will be a bias in this approach. Please comment.

Methods, line 317: This sentence should be removed or shifted to the Discussion. This was not performed in their study but is planned for future studies.

Methods, line 322: Were the MRS voxel positions considered for biopsy sampling. How was biopsy sampling performed? Please clarify.

Methods, line 323: Which method was used for snap freezing?

Results, Figure 2: The two spectra show strong differences in the lactate resonance. Did the authors perform an analysis in the region around 1.3 ppm too?

Results, line 154: The authors should mention that ex vivo analysis was performed in eight patients only.

Results, line 159: The results for the comparison of in vivo MRS and ex vivo NMRS data are sparse. The authors should provide a Figure comparing ex vivo NMR spectra from IDHwt and IDHmut glioma with the respective in vivo MRS spectra.

Discussion, line 167: I’m not sure that this really is a prospective study.

Discussion, line 174: The authors did not present discriminating features in ex vivo NMRS data. Please remove this statement or include these results (as suggested above).

Discussion, line 193: What is meant with “… an average twenty minutes additional examination period following standard MRI”? I guess the MRS needed 3:30 minutes for data acquisition. Please clarify.

Discussion, line 228: The authors should include more limitations of their study, as suggested above.

Reviewer 2 Report

In their manuscript, the authors report their development of a lSVM classifier for predicting IDH mutational status from 1H MRS spectra. IDH mutational status defines two biologically distinct diseases and its detection is therefore of high clincal relevance.

There are a couple of points which must be addressed:

1) As expected, IDH status also separates LGG (II/III) and GBM (IV). Is your classifier truly detecting IDH, or the WHO grade as a surrogate marker? This should be investigated. Please also provide (maybe in Supplement) prediction results for all samples.

2) There is a plethora of literature (e.g. http://www.ajnr.org/content/early/2018/05/10/ajnr.A5667) on using for example anatomic images (T1(c), T2, FLAIR) for predicting IDH, which works comparably well to your approach and is validated in much larger cohorts such as BraTS. What is the benefit of your approach which requires MRS?

3) A more general question: Do you think your test could substitute for a biopsy? Otherwise, what is the clinical motivation for predicting IDH? What about 1p/19q, MGMT,...?

Round 2

Reviewer 1 Report

No further corrections are required. The authors adequately answered all of my questions.

Reviewer 2 Report

Thank you for answering my questions.